# Sparse Dictionary Learning by Dynamical Neural Networks

**Tsung-Han Lin, Ping Tak Peter Tang**
Intel Corporation
Santa Clara, CA
`{tsung-han.lin,peter.tang}@intel.com`

## Abstract

A *dynamical neural network* consists of a set of interconnected neurons that interact over time continuously. It can exhibit computational properties in the sense that the dynamical system's evolution and/or limit points in the associated state space can correspond to numerical solutions to certain mathematical optimization or learning problems. Such a computational system is particularly attractive in that it can be mapped to a massively parallel computer architecture for power and throughput efficiency, especially if each neuron can rely solely on local information (i.e., local memory). Deriving gradients from the dynamical network's various states while conforming to this last constraint, however, is challenging. We show that by combining ideas of *top-down feedback* and *contrastive learning*, a dynamical network for solving the $\ell_1$-minimizing dictionary learning problem can be constructed, and the true gradients for learning are provably computable by individual neurons. Using spiking neurons to construct our dynamical network, we present a learning process, its rigorous mathematical analysis, and numerical results on several dictionary learning problems.

## 1 Introduction

A network of simple neural units can form a physical system that exhibits computational properties. Notable examples include Hopfield network (Hopfield, 1982) and Boltzmann machine (Ackley et al., 1985). Such systems have global states that evolve over time through only *local interactions* among neural units. Typically, one is interested in a system whose motion converges towards locally stable limit points, with the limit points representing the computational objective of interest. For example, a Hopfield network's limit points correspond to stored memory information and that of a Boltzmann machine, a data representation. These computational systems are interesting for both engineering and neuroscience research. From a hardware implementation standpoint, such computational models allow the mapping of neurons to a massively parallel architecture (Davies et al., 2018; Merolla et al., 2014). By allocating private local memory to each processing element, the so-called von Neumann memory bottleneck in modern computers can be eliminated, delivering much greater power and throughput efficiency (e.g., see Kung (1982)). For neuroscience, such computational models obey the fundamental physical locality constraints of biological neurons, providing a direction for understanding the brain.

We are interested in using such systems to solve the $\ell_1$-minimizing sparse coding and dictionary learning problem, which has fundamental importance in many areas, e.g., see Mairal et al. (2014). It is well-known that even just the sparse coding problem, with a prescribed dictionary, is non-trivial to solve, mainly due to the non-smooth objective involving an $\ell_1$-norm (Efron et al., 2004; Beck & Teboulle, 2009). Remarkably, a dynamical network known as the LCA network (Rozell et al., 2008) can be carefully constructed so that its limit points are identical to the solution of the sparse coding problem. Use of a dynamical network thus provides an alternative and potentially more power efficient method for sparse coding to standard numerical optimization techniques. Nevertheless, while extending numerical optimization algorithms to also learning the underlying dictionary is somewhat straightforward, there is very little understanding in using dynamical networks to learn a dictionary with provable guarantees due to the challenging locality constraints.

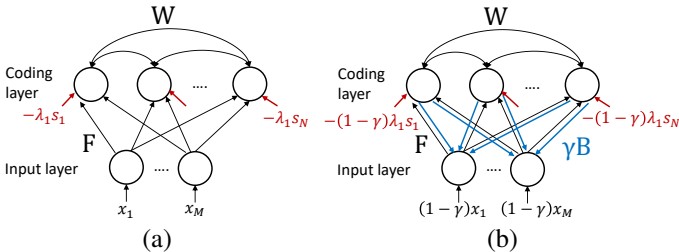

Figure 1: The network topologies discussed in this work. (a) is known as the LCA network that can perform sparse coding. We propose the network in (b) for dictionary learning.

In this work, we devise a new network topology and learning rules that enable dictionary learning. In particular, we show that the gradients for learning are provably computable by individual neurons using only local information. On a high level, our learning strategy is similar to the contrastive learning procedure developed in training Boltzmann machines, which also gathers much recent interest in deriving implementations of backpropagation under the same neuron locality constraints (Ackley et al., 1985; Movellan, 1990; O'Reilly, 1996; Xie & Seung, 2003; Scellier & Bengio, 2017; Whittington & Bogacz, 2017). During training, the network is run in two different configurations – a "normal" one and a "perturbed" one.[1] The networks' limit points under these two configurations will be identical if the weights to be trained are already optimal, but different otherwise. The learning process is a scheme to so adjust the weights to minimize the difference in the limit points. In Boltzmann machine, the weight adjustment can be formulated as minimizing a KL divergence objective function.

For dictionary learning, we adopt a neuron model whose activation function corresponds to the unbounded ReLU function rather than the bounded sigmoid-like function in Hopfield networks or Boltzmann machines, and a special network topology where connection weights have dependency. Interestingly, the learning processes are still similar: We also rely on running our network in two configurations. The difference in states after a long-enough evolution, called limiting states in short, is shown to hold the gradient information of a dictionary learning objective function which the network minimizes, as well as the gradient information for the network to maintain weight dependency. Comparisons between this work, Hopfield network, and Boltzmann machine can be found in Appendix C.1.

## 1.1 Related Work

Dictionary learning is thought to be related to the formation of receptive fields in visual cortex (Olshausen & Field, 1996) and has been widely studied. The typical architecture studied is a feedforward-only, two-layer neural network with inhibitory lateral connections among the second layer neurons as shown in Figure 1(a) (Földiak, 1990; Zylberberg et al., 2011; Brito & Gerstner, 2016; Hu et al., 2014; Seung & Zung, 2017; Pehlevan et al., 2018; Vertechi et al., 2014; Brendel et al., 2017). The lateral connections allow the coding neurons to compete among themselves and hence induce sparseness in neural activities, giving dynamics more complex than the conventional deep neural networks, which do not have intra-layer connections.[2] In Rozell et al. (2008), it is shown that the coding neuron activations can correspond to a sparse coding solution if the connection weights are set according to a global dictionary $D$ as $F = D^T$, $W = -D^T D + I$.[3] To enable learning in this network (that is, each neuron locally adjusts their connection weights to adapt the dictionary; see Section 2.2 for the definition of weight locality), one must address the following two questions:

- How does individual neuron compute the gradient for learning locally?
- How do the neurons collectively maintain the global weight consistency between $F$ and $W$?

---

[1] In Botlzmann machine, the two configurations are called the *free-running phase* and the *clamped phase*.

[2] This should not be confused with the conventional recurrent neural networks. Although RNNs also have intra-layer connections, these connections are still uni-directional over a sequence of input.

[3] The exact formulation depends on the neuron model. In the spiking neuron formulation, we in fact have $W - \Theta = -D^T D$ where $\Theta$ is the firing thresholds. See Section 3.1 for more details.

The first line of work, Földiak (1990); Zylberberg et al. (2011); Brito & Gerstner (2016), adopts the Hebbian/anti-Hebbian heuristics for learning the feedforward and lateral weights, respectively, and empirically demonstrated that such learning yielded Gabor-like receptive fields if trained with natural images. However, unlike the network in Rozell et al. (2008), this learning heuristic is not mathematically derived from a rigorous learning objective, and hence cannot address any of the two above questions. Recently, it is shown that learning rules resembling the Hebbian/anti-Hebbian heuristic can be derived from minimizing a "similarity matching" objective function between input and output correlations (Hu et al., 2014; Pehlevan et al., 2018). This formulation is somewhat different from the common autoencoder-style dictionary learning formulation discussed in this work.

Another line of work, Vertechi et al. (2014); Brendel et al. (2017), notes the importance of *balance* between excitation and inhibition among the coding neurons, and proposes that the learning target of lateral connections should be to maintain such balance. That is, the inhibitory lateral weights should grow according to the feedforward excitation, and hence potentially can ensure the weight consistency between $F$ and $W$. Nevertheless, similar to the first line of work, both Vertechi et al. (2014); Brendel et al. (2017) resort to pure Hebbian rule when learning the feedforward weights $F$ (or equivalently, learning the dictionary). Since the Hebbian rule does not necessarily follow a descending direction that minimizes the dictionary learning objective function, the convergence of this learning approach to a local minimum still cannot be guaranteed. Further discussions of prior work are provided in Appendix C.2.

## 1.2 Contributions

The major advance in this work is to recognize the inadequacy of the customary feedforward-only architecture, and to introduce *top-down feedback* connections shown in Figure 1(b). As will later be shown, this network structure allows the true learning gradients to be provably computable from the resulting network dynamics. Further, the existence of feedback allows us to devise a separate mechanism that acts as an inner loop during learning to continuously ensure weight consistency among all connections. Combining these two, we can successfully address both the above questions and the dictionary learning problem.

We will focus our discussion on a network that uses *spiking neurons* as the basic units that are suited for digital circuit implementations with high computational efficiency. Note that this does not result in a loss of generality. The principles of LCA network can be applied to both continuous-valued and spiking neurons (Shapero et al., 2014; Tang et al., 2017), and similarly the results established in this paper can be easily applied to construct a network of continuous-valued neurons for dictionary learning.

Finally, we note that in parallel to this work, various forms of feedback connections and mechanisms have been proposed to address neural network local learning in the context of other learning problems, including autoencoder networks (Hinton & McClelland, 1988; Burbank, 2015), deep neural networks (Guerguiev et al., 2017; Sacramento et al., 2018), and memory networks (Federer & Zylberberg, 2018). This work further shows that feedback connection is also a crucial component for solving the complex dictionary learning problem.

## 2 Background

### 2.1 Integrate-and-Fire Spiking Neuron Model and Network Dynamics

An integrate-and-fire neuron has two internal state variables that govern its dynamics: the *current* $\mu(t)$ and the *potential* $\rho(t)$. The key output of a neuron is a time sequence of spikes – spike train – that it produces. A neuron's spike train is generated by its potential $\rho(t)$; $\rho(t)$ is in turn driven by the current $\mu(t)$, which is in turn driven by a *constant bias* $\beta$ (bias in short) and the spike trains of other neurons to which it is connected. Specifically, each neuron has a configured firing threshold $\theta > 0$. When $\rho(t)$ reaches $\theta$, say at time $t_k$, a spike given by the Dirac delta function $\delta(t - t_k)$ is generated and $\rho(t)$ is reset to 0: $\rho(t_k^+) = 0$. For $t > t_k$ and before $\rho(t)$ reaches $\theta$ again, $\rho(t) = \int_{t_k}^t \mu(s)\,ds$.

In a system of $N$ neurons $n_i$, $i = 1, 2, \ldots, N$, let $\sigma_j(t) = \sum_k \delta(t - t_{j,k})$ denote the spike train of neuron $n_j$. The current $\mu_i(t)$ of $n_i$ is given in terms of its bias $\beta_i$ and the spike trains $\{\sigma_j(t)\}$:

$$\mu_i(t) = \beta_i + \sum_{j \neq i} W_{ij} \, (\alpha * \sigma_j)(t), \tag{1}$$

where $\alpha(t) = \frac{1}{\tau} e^{-t/\tau}$ for $t \geq 0$, $\alpha(t) = 0$ for $t < 0$ and $*$ is the convolution operator. Neuron $n_j$ inhibits (excites) $n_i$ if $W_{ij} < 0$ ($W_{ij} > 0$). If $W_{ij} = 0$, neurons $n_i$ and $n_j$ are not connected. For simplicity, we consider only $\tau = 1$ throughout the paper. Equation 1 yields the dynamics

$$\dot{\boldsymbol{\mu}}(t) = \boldsymbol{\beta} - \boldsymbol{\mu}(t) + W \cdot \boldsymbol{\sigma}(t), \tag{2}$$

where the vectors $\boldsymbol{\mu}(t)$ and $\boldsymbol{\sigma}(t)$ denote the $N$ currents and spike trains (see Appendix B.1 for the full derivation.)

The network dynamics can be studied via the filtered quantities of average current and spike rate:

$$\mathbf{u}(t) \stackrel{\text{def}}{=} \frac{1}{t} \int_0^t \boldsymbol{\mu}(s) \, ds, \qquad \mathbf{a}(t) \stackrel{\text{def}}{=} \frac{1}{t} \int_0^t \boldsymbol{\sigma}(s) \, ds. \tag{3}$$

In terms of $\mathbf{u}(t)$ and $\mathbf{a}(t)$, Equation 2 becomes

$$\dot{\mathbf{u}}(t) = \boldsymbol{\beta} - \mathbf{u}(t) + W \mathbf{a}(t) + (\boldsymbol{\mu}(0) - \mathbf{u}(t))/t \tag{4}$$

The trajectory $(\mathbf{u}(t), \mathbf{a}(t))$ has interesting properties. In particular, Theorem 1 below (cf. Tang et al. (2017)) shows that any limit point $(\mathbf{u}^*, \mathbf{a}^*)$ satisfies $\mathbf{u}^* - \Theta \mathbf{a}^* \leq \mathbf{0}$, $\mathbf{a}^* \geq \mathbf{0}$ and $(\mathbf{u}^* - \Theta \mathbf{a}^*) \odot \mathbf{a}^* = \mathbf{0}$ where $\odot$ is elementwise product. These properties are crucial to Section 3.

**Theorem 1.** *Let* $\Theta = \mathrm{diag}(\boldsymbol{\theta})$, $\boldsymbol{\theta} = [\theta_1, \theta_2, \ldots, \theta_N]$, *then*

$$\mathbf{u}(t) - \Theta \mathbf{a}(t) = \boldsymbol{\beta} + (W - \Theta) \cdot \mathbf{a}(t) + \boldsymbol{\Delta}(t) \tag{5}$$

*where* $\max(\mathbf{u}(t), \mathbf{0}) - \Theta \mathbf{a}(t) \to \mathbf{0}$ *and* $\boldsymbol{\Delta}(t) \to \mathbf{0}$.

As with all other theorems, Theorem 1 is given in a conceptual form where the corresponding rigorous "$\epsilon$-$\delta$" versions are detailed in the Appendix.

## 2.2 PARALLEL MODEL OF DYNAMICAL NEURAL NETWORKS

We view the dynamical network as a computational model where each neuron evolves in parallel and asynchronously. One-sided communication in the form of a one-bit signal from Neuron $n_j$ to Neuron $n_i$ occurs only if the two are connected and only when the former spikes. The network therefore can be mapped to a massively parallel architecture, such as Davies et al. (2018), where the connection weights are stored distributively in each processing element's (PE) local memory. In the most general case, we assume the architecture has the same number of PEs and neurons; each PE hosts one neuron and stores the weights connected towards this neuron, that is, each PE stores one row of the $W$ matrix in Equation 2. With proper interconnects among PEs to deliver spike messages, the dynamical network can be realized to compute sparse coding solutions.

This architectural model imposes a critical weight locality constraint on learning algorithms for dynamical networks: The connection weights must be adjusted with rules that rely only on locally available information such as connection weights, a neuron's internal states, and the rate of spikes it receives. The goal of this paper is to enable dictionary learning under this locality constraint.

## 3 DICTIONARY LEARNING

In dictionary learning, we are given $P$ images $\mathbf{x}^{(p)} \in \mathbb{R}_{\geq 0}^M$, $p = 1, 2, \ldots, P$. The goal is to find a dictionary consisting of a prescribed number of $N$ atoms, $D = [\mathbf{d}_1, \mathbf{d}_2, \ldots, \mathbf{d}_N]$, $D \in \mathbb{R}^{M \times N}$ such that each of the $P$ images can be sparsely coded in $D$. We focus here on non-negative dictionary and formulate our minimization problem as

$$\arg\min_{\mathbf{a}^{(p)} \geq \mathbf{0}, D \geq \mathbf{0}} \sum_{p=1}^P l(D, \mathbf{x}^{(p)}, \mathbf{a}^{(p)}), \quad l(D, \mathbf{x}, \mathbf{a}) = \frac{1}{2} \|\mathbf{x} - D\mathbf{a}\|_2^2 + \lambda_1 \|S\mathbf{a}\|_1 + \frac{\lambda_2}{2} \|D\|_F^2, \tag{6}$$

$S$ being a positive diagonal scaling matrix.

Computational methods such as stochastic online training (Aharon & Elad, 2008) is known to be effective for dictionary learning. With this method, one iterates on the following two steps, starting with a random dictionary.

1. Pick a random image $\mathbf{x} \leftarrow \mathbf{x}^{(p)}$ and obtain sparse code $\mathbf{a}$ for the current dictionary $D$ and image $\mathbf{x}$, that is, solve Equation (6) with $D$ fixed.

2. Use gradient descent to update $D$ with a learning rate $\eta$. The gradient $\nabla_D$ with respect to $D$ is in a simple form and the update of $D$ is

$$D^{(\text{new})} \leftarrow D - \eta \left( (D\mathbf{a} - \mathbf{x})\mathbf{a}^T + \lambda_2 D \right). \tag{7}$$

Implementing these steps with a dynamical network is challenging. First, previous works have only shown that Step 1 can be solved when the configuration uses the dictionary $D$ in the feedforward connection weights and $D^T D$ as the lateral connection weights (Shapero et al. (2014), c.f. Figure 1(a) and below). For dictionary learning, both sets of weights evolve without maintaining this exact relationship, casting doubt if Step 1 can be solved at all. Second, the network in Figure 1(a) only has $F = D^T$, rendering the needed term $D\mathbf{a}$ uncomputable using information local to each neuron. Note that in general, gradients to minimize certain objective functions in a neural network can be mathematically derived, but often times they cannot be computed locally, e.g., standard backpropagation and general gradient calculations for spiking networks (Huh & Sejnowski, 2017). We now show that our design depicted in Figure 1(b) can indeed implement Steps 1 and 2 and solve dictionary learning.

### 3.1 SPARSE CODING – GETTING $\mathbf{a}$

Non-negative sparse coding (Equation 6 with $D$ fixed) is a constrained optimization problem. The standard approach (cf. Boyd & Vandenberghe (2004)) is to augment $l(D, \mathbf{x}, \mathbf{a})$ with non-negative slack variables, with which the optimal solutions are characterized by the KKT conditions. Consider now Figure 1(b) that has explicit feedback weights $B$ whose strength is controlled by a parameter $\gamma$. Equation 5, reflecting the structure of the coding and input neurons, takes the form:

$$\begin{bmatrix} \mathbf{e}_\gamma(t) \\ \mathbf{f}_\gamma(t) \end{bmatrix} \stackrel{\text{def}}{=} \begin{bmatrix} \mathbf{u}_\gamma(t) - \Theta\mathbf{a}_\gamma(t) \\ \mathbf{v}_\gamma(t) - \mathbf{b}_\gamma(t) \end{bmatrix} = \begin{bmatrix} -(1-\gamma)\lambda_1\mathbf{s} \\ (1-\gamma)\mathbf{x} \end{bmatrix} + \begin{bmatrix} -H & F \\ \gamma B & -I \end{bmatrix} \begin{bmatrix} \mathbf{a}_\gamma(t) \\ \mathbf{b}_\gamma(t) \end{bmatrix} + \mathbf{\Delta}(t) \tag{8}$$

$(\mathbf{u}(t), \mathbf{v}(t))$ and $(\mathbf{a}(t), \mathbf{b}(t))$ denote the average currents and spike rates for the coding and input neurons, respectively, and $H \stackrel{\text{def}}{=} W + \Theta$. Note that when $\gamma = 0$, the network is equivalent to Figure 1(a). It is established in Tang et al. (2017) that when $F^T = D$, $H = D^T D$ and at a limit point $(\mathbf{e}_0^*, \mathbf{a}_0^*)$, Equation 8 is simplified and reduces to $\mathbf{e}_0^* = -\lambda_1\mathbf{s} - D^T D\mathbf{a}_0^* + D^T\mathbf{x}$ and that $\mathbf{e}_0^* \leq \mathbf{0}$, $\mathbf{a}_0^* \geq \mathbf{0}$ and $\mathbf{e}_0^* \odot \mathbf{a}_0^* = \mathbf{0}$. This shows that $\mathbf{a}_0^*$ and $-\mathbf{e}_0^*$ are the optimal primal and slack variables that satisfy the KKT conditions. In particular $\mathbf{a}_0^*$ is the optimal sparse code.

For simplicity, we consider in this section that the feed-forward and feedback weights in Figure 1(b) are initialized to be symmetric and correspond to a global dictionary $D$, $F^T = B = D$ (see further discussions in Section 3.3 for the general case.) This network can similarly solve the sparse coding problem. We extend the previously established result (Tang et al., 2017) in several aspects: (1) $\gamma$ can be set to any values in $[0, 1)$; all $\mathbf{a}_\gamma^*$ are the optimal sparse code, (2) $H$ needs not be $FB$ exactly; $\|H - FB\|$ being small suffices, and (3) as long as $t$ is large enough, $\mathbf{a}_\gamma(t)$ solves an approximate sparse coding problem. These are summarized as follows (where the rigorous form is presented in the Appendix).

**Theorem 2.** *Let $F^T = B = D$, $\gamma \in [0, 1)$ and $\|H - FB\|$ be small. Then for $t$ large enough, $\mathbf{a}_\gamma(t)$ is close to an exact solution $\tilde{\mathbf{a}}$ to Equation 6 ($D$ fixed) with $S$ replaced by $\tilde{S}$ where $\|S - \tilde{S}\|$ is small.*

The significant implication is that despite slight discrepancies between $H$ and $FB$, the average spike rate $\mathbf{a}_\gamma(t)$ at $t$ large enough is a practical solution to Step 1 of the stochastic learning procedure.

## 3.2 Dictionary Adjustment – Updating $F, B$ and $H$

To obtain the learning gradients, we run the network for a long enough time to sparse code twice: at $\gamma = 0$ and $\gamma = \kappa > 0$, obtaining $\tilde{\mathbf{e}}_0, \tilde{\mathbf{e}}_\kappa, \tilde{\mathbf{a}}_0, \tilde{\mathbf{a}}_\kappa$ and $\tilde{\mathbf{b}}_0, \tilde{\mathbf{b}}_\kappa$ at those two configurations. We use tilde to denote the obtained states and loosely call them as limiting states. Denote $1 - \kappa$ by $\kappa^c$.

**Theorem 3.** *The limiting states satisfy*

$$\kappa(B\tilde{\mathbf{a}}_\kappa - \mathbf{x}) \approx \mathbf{g}_D, \qquad \mathbf{g}_D \stackrel{\text{def}}{=} \tilde{\mathbf{b}}_\kappa - \tilde{\mathbf{b}}_0 \tag{9}$$

$$\kappa(H - FB)\tilde{\mathbf{a}}_\kappa \approx \mathbf{g}_H, \qquad \mathbf{g}_H \stackrel{\text{def}}{=} \kappa^c H(\tilde{\mathbf{a}}_0 - \tilde{\mathbf{a}}_\kappa) + (\kappa^c \tilde{\mathbf{e}}_0 - \tilde{\mathbf{e}}_\kappa) \tag{10}$$

We now show Theorem 3 lays the foundation for computing all the necessary gradients that we need. Equation 9 shows that (recall $B = D$)

$$D\tilde{\mathbf{a}}_\kappa - \mathbf{x} \approx \kappa^{-1}\mathbf{g}_D.$$

In other words, the spike rate differences at the input layer reflect the reconstruction error of the sparse code we just computed. Following Equation 7, this implies that the update to each weight can be approximated from the spike rates of the two neurons that it connects, while the two spike rates surely are locally available to the destination neuron that stores the weight. Specifically, each coding neuron has a row of the matrix $F = D^T$; each input neuron has a row of the matrix $B = D$. These neurons each updates its row of matrix via

$$\begin{aligned} F_{ij}^{(\text{new})} &\leftarrow F_{ij} - \eta_D \left( \kappa^{-1}(\tilde{\mathbf{a}}_\kappa)_i (\mathbf{g}_D)_j + \lambda_2 F_{ij} \right) \\ B_{ij}^{(\text{new})} &\leftarrow B_{ij} - \eta_D \left( \kappa^{-1}(\tilde{\mathbf{a}}_\kappa)_j (\mathbf{g}_D)_i + \lambda_2 B_{ij} \right) \end{aligned} \tag{11}$$

Note that $F^T = B = D$ is maintained.

Ideally, at this point the $W$ and $\Theta$ stored distributively in the coding neurons will be updated to $H^{(\text{new})}$ where $H^{(\text{new})} = F^{(\text{new})} B^{(\text{new})}$. Unfortunately, each coding neuron only possesses one row of the matrix $F^{(\text{new})}$ and does not have access to any values of the matrix $B^{(\text{new})}$. To maintain $H$ to be close to $D^T D$ throughout the learning process, we do the following. First we aim to modify $H$ to be closer to $FB$ (not $F^{(\text{new})} B^{(\text{new})}$) by reducing the cost function $\phi(H) = \frac{1}{2}\|(H - FB)\tilde{\mathbf{a}}_\kappa\|_2^2$. The gradient of this cost function is $\nabla_H \phi = (H - FB)\tilde{\mathbf{a}}_\kappa \tilde{\mathbf{a}}_\kappa^T$ which is computable as follows. Equation 10 shows that

$$\nabla_H \phi \approx G \stackrel{\text{def}}{=} \kappa^{-1} \mathbf{g}_H \tilde{\mathbf{a}}_\kappa^T$$

Using this approximation, coding neuron $n_{C,i}$ has the information to compute the $i$-th row of $G$. We modify $H$ by $-\eta_H G$ where $\eta_H$ is some learning rate. This modification can be thought of as a catch-up correction because $F$ and $B$ correspond to the updated values from a previous iteration. Because the magnitude of that update is of the order of $\eta_D$, we have $\|H - FB\| \approx \eta_D$ and $\|G\| \approx \eta_D$. Thus $\eta_H$ should be bigger than $\eta_D$ lest $\|\eta_H G\| \approx \eta_H \eta_D$ be too small to be an effective correction. In practice, $\eta_H \approx 15\eta_D$ works very well.

In addition to this catch-up correction, we also make correction of $H$ due to the update of $-\eta_D \lambda_2 F$ and $-\eta_D \lambda_2 B$ to $F$ and $B$. These updates lead to a change of $-2\eta_D FB + O(\eta_D^2)$. Consequently, after Equation 11, we update $H$ by

$$H_{ij}^{(\text{new})} \leftarrow H_{ij} - \eta_H \kappa^{-1}(\mathbf{g}_H)_i(\mathbf{a}_\kappa)_j - 2\eta_D \lambda_2 H_{ij}. \tag{12}$$

Note that the update to $H$ involves update to the weights $W$ as well as the thresholds $\Theta$ (recall that $H \stackrel{\text{def}}{=} W + \Theta$). Combining the above, we summarize the full dictionary learning algorithm below.

## 3.3 Discussions

**Dictionary norm regularization.** In dictionary learning, typically one needs to control the norms of atoms to prevent them from growing arbitrarily large. The most common approach is to constrain the atoms to be exactly (or at most) of unit norms, achieved by re-normalizing each atom after a dictionary update. This method however cannot be directly adopted in our distributed setting. Each input neuron only has a row of the matrix $B$ but not a column of $B$ – an atom – so as to re-normalize.

---

**Algorithm 1** Dictionary Learning

---

*Initialization:* Pick a random dictionary $D \geq \mathbf{0}$ with atoms of unit Euclidean norm. Configure $F \leftarrow D^T$, $B \leftarrow D$, $\mathbf{s} \leftarrow [1, 1, \ldots, 1]^T$, and $H \leftarrow FB$.
**repeat**
    1. *Online input:* Pick a random image $\mathbf{x}$ from $\{\mathbf{x}^{(p)}\}$
    2. *Sparse coding:* Run the network at $\gamma \leftarrow 0$ and at $\gamma \leftarrow \kappa > 0$.
    3. *Dictionary update:* Compute the vectors $\mathbf{g}_D$ and $\mathbf{g}_H$ distributively according to Equations 9 and 10. Update $F$, $B$ and $H$ according to Equations 11 and 12. Project the weights to non-negative quadrant.
    4. *Scaling update:* Set the scaling vector $\mathbf{s}$ to $\mathrm{diag}(H)$. This scaling helps maintain each atom of the dictionary to be of similar norms.
**until** dictionary is deemed satisfactory

---

We chose instead to regularize the Frobenius norm of the dictionaries, translating to a simple decay term in the learning rules. This regularization alone may result in learning degenerate zero-norm atoms because sparse coding tends to favor larger-norm atoms to be actively updated, leaving smaller-norm ones subject solely to continual weight decays. By choosing a scaling factor $\mathbf{s}$ set to $\mathrm{diag}(H)$, sparse coding favors smaller-norm atoms to be active and effectively mitigates the problem of degeneracy.

**Boundedness of network activities.** Our proposed network is a feedback nonlinear system, and one may wonder whether the network activities will remain bounded. While we cannot yet rigorously guarantee boundedness and stability under some a priori conditions, currents and spike rates remain bounded throughout learning for all our experiments. One observation is that the feedback excitation amounts to $\gamma FB\mathbf{a}_\gamma(t)$ and the inhibition is $H\mathbf{a}_\gamma(t)$. Therefore when $H = FB$ and $\gamma < 1$, the feedback excitation is nullified, keeping the network from growing out of bound.

**Network execution in practice.** Theoretically, an accurate spike rate can only be measured at a very large $T$ as precision increases at a rate of $O(1/t)$. In practice, we observed that a small $T$ suffices for dictionary learning purpose. Stochastic gradient descent is known to be very robust against noise and thus can tolerate the low-precision spike rates as well as the approximate sparse codes due to the imperfect $H \approx FB$. For faster network convergence, the second network $\gamma = \kappa$ is ran right after the first network $\gamma = 0$ with all neuron states preserved.

**Weight symmetry.** The sparse code and dictionary gradient are computed using the feedforward and feedback weights respectively. Therefore a symmetry between those weights is the most effective for credit assignment. We have assumed such symmetry is initialized and the learning rules can subsequently maintain the symmetry. One interesting observation is that even if the weights are asymmetric, our learning rules still will symmetrize them. Let $E_{ij}^{(p)} = F_{ji}^{(p)} - B_{ij}^{(p)}$ be the weight difference at the $p$-th iteration. It is straightforward to show $E_{ij}^{(p)} = \alpha^{p-1} E_{ij}^{(1)}$, $\alpha = 1 - \eta_D \lambda_2$. Hence $E_{ij}^{(p)} \to 0$ as $p$ gets bigger. In training deep neural networks, symmetric feedforward and feedback weights are important for similar reasons. The lack of local mechanisms for the symmetry to emerge makes backpropagation biologically implausible and hardware unfriendly, see for example Liao et al. (2016) for more discussions. Our learning model may serve as a building block for the pursuit of biologically plausible deep networks with backpropagation-style learning.

## 4 NUMERICAL EXPERIMENTS

We examined the proposed learning algorithm using three datasets. **Dataset A.** 100K randomly sampled $8 \times 8$ patches from the grayscale Lena image to learn 256 atoms. **Dataset B.** 50K $28 \times 28$ MNIST images (LeCun et al., 1998) to learn 512 atoms. **Dataset C.** 200K randomly sampled $16 \times 16$ patches from whitened natural scenes (Olshausen & Field, 1996) to learn 1024 atoms. These are standard datasets in image processing (A), machine learning (B), and computational neuroscience (C).[4] For each input, the network is ran with $\gamma = 0$ from $t = 0$ to $t = 20$ and with $\gamma = 0.7$ from $t = 20$ to $t = 40$, both with a discrete time step of $1/32$. Note that although this time window of 20

---

[4]For Dataset A and C, the patches are further subtracted by the means, normalized, and split into positive and negative channels to create non-negative inputs (Hoyer, 2004).

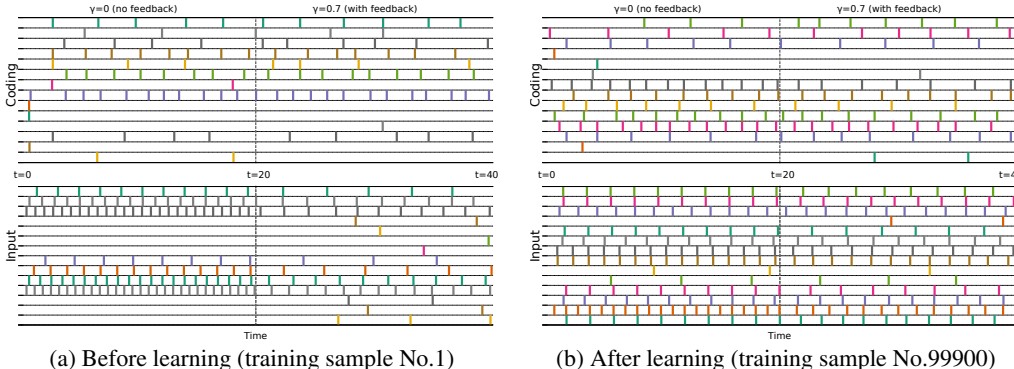

(a) Before learning (training sample No.1)  (b) After learning (training sample No.99900)

Figure 2: Network spike patterns in practice. In the figures, each row corresponds to one neuron, and the bars indicate the spike timings. Following Algorithm 1, for each online input the network is run with $\gamma = 0$ from $t = 0$ to $t = 20$ (feed-forward only), and $\gamma = 0.7$ from $t = 20$ to $t = 40$ (perturb network activities by feedback). The spike rates $\tilde{\mathbf{a}}$, $\tilde{\mathbf{b}}$ for computing the gradients are then collected by counting the number of spikes within the time interval. The figures in the left and right show the spike patterns before and after learning, respectively. We make two observations: (1) The coding neuron spike rates remain approximately constant for both intervals $t = [0, 20]$ and $t = [20, 40]$, illustrating Theorem 2 that feedback perturbation does not alter the computed optimal sparse code. This property holds throughout learning because our network learns to maintain weight consistency. (2) For the figure on the left (before learning with a random dictionary), the input neuron spike rates change significantly when feedback exists. From Theorem 3, this spike rate difference encodes the reconstruction error of the computed sparse code. This quantity is used for gradient computation and is expected to be minimized according to the learning objective. Indeed, after learning (figure on the right), the input neuron spike rates exhibit much less perturbation by the feedback. This shows the network is able to learn a proper dictionary that minimizes reconstruction error. Data is from learning with Dataset A; only a subset of the neurons are shown.

is relatively small and yields a spike rate precision of only 0.05, we observed that it is sufficient for gradient calculation and dictionary learning purpose.

We explored two different connection weight initialization schemes. First, we initialize the weights to be fully consistent with respect to a random dictionary. Second, we initialized the weights to be asymmetric. In this case, we set $F^T$ and $B$ to be column-normalized random matrices and the entries of $H$ to be random values between [0, 1.5] with the diagonal set to 1.5.

## 4.1 Network Dynamics

We first show the spike patterns from a network with fully consistent initial weights in Figure 2. It can be seen that the spike patterns quickly settle into a steady state, indicating that a small time window may suffice for spike rate calculations. Further, we can observe that feedback only perturbs the input neuron spike rates while keeping the coding neuron spike rates approximately the same, validating our results in Section 3.1 and 3.2.

Another target the algorithm aims at is to approximately maintain the weight consistency $H \approx FB$ during learning. Figure 3 shows that this is indeed the case. Note that our learning rule acts as a catch-up correction, and so an exact consistency cannot be achieved. An interesting observation is that as learning proceeds, weight consistency becomes easier to maintain as the dictionary gradually converges.

Although we have limited theoretical understanding for networks with random initial weights, Figure 3 shows that our learning procedure can automatically discover consistent and symmetric weights with respect to a single global dictionary. This is especially interesting given that the neurons only learn with local information. No neuron has a global picture of the network weights.

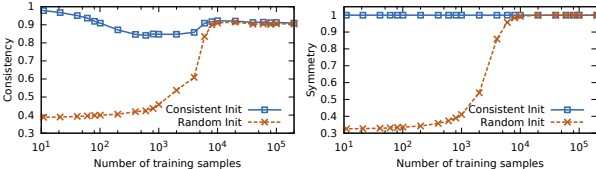

Figure 3: Network weight consistency and symmetry during learning. Consistency is measured as $1 - \|H - FB\|_F / \|H\|_F$. Symmetry is measured as the average normalized inner product between the $i$-th row of $F$ and the $i$-th column of $B$ for $i = 1 \ldots N$. Data is from learning with Dataset A.

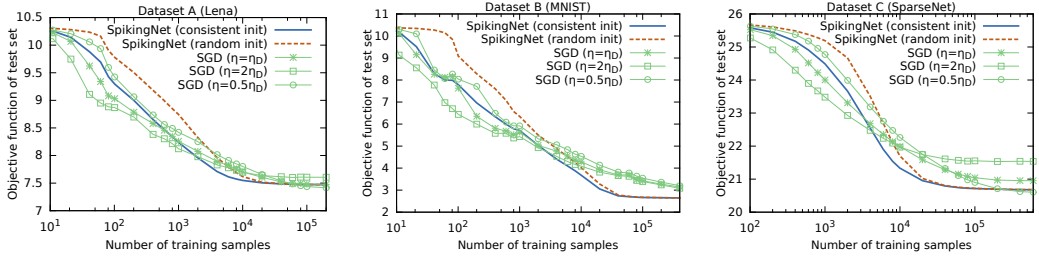

Figure 4: Comparison of convergence of learning with dynamical neural network and SGD.

## 4.2 Convergence of Dictionary Learning

The learning problem is non-convex, and hence it is important that our proposed algorithm can find a satisfying local minimum. We compare the convergence of spiking networks with the standard stochastic gradient descent (SGD) method with the unit atom norm constraint. For simplicity, both algorithms use a batch size of 1 for gradient calculations. The quality of the learned dictionary $D = F^T$ is measured using a separate test set of 10K samples to calculate a surrogate dictionary learning objective (Mairal et al., 2009). For a fair comparison, the weight decay parameters in spiking networks are chosen so that the average atom norms converge to approximately one.

Figure 4 shows that our algorithm indeed converges and can obtain a solution of similar, if not better, objective function values to SGD consistently across the datasets. Surprisingly, our algorithm can even reach a better solution with fewer training samples, while SGD can be stuck at a poor local minimum especially when the dictionary is large. This can be attributed to the $\ell_1$-norm reweighting heuristic that encourages more dictionary atoms to be actively updated during learning. Finally, we observe that a network initialized with random non-symmetric weights still manages to reach objective function values comparable to those initialized with symmetric weights, albeit with slower convergence due to less accurate gradients. From Figure 3, we see the network weights are not symmetric before $10^4$ samples for Dataset A. On the other hand, from Figure 4 the network can already improve the dictionary before $10^4$ samples, showing that perfectly symmetric weights are not necessary for learning to proceed.

## 5 Conclusion

We have presented a dynamical neural network formulation that can learn dictionaries for sparse representations. Our work represents a significant step forward that it not only provides a link between the well-established dictionary learning problem and dynamical neural networks, but also demonstrates the contrastive learning approach to be a fruitful direction. We believe there is still much to be explored in dynamical neural networks. In particular, learning in such networks respects data locality and therefore has the unique potential, especially with spiking neurons, to enable low-power, high-throughput training with massively parallel architectures.

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

# Appendices

## A   DETAILED DESCRIPTION OF PROPOSED NETWORK STRUCTURE

We propose a novel network topology with feedback shown in Figure 1(b). The figure shows two "layers" of neurons. The lower layer consists of $M$ neurons we call input neurons, $n_{I,i}$ for $i = 1, 2, \ldots, M$; the upper layer consists of $N$ neurons we call coding neurons $n_{C,i}$ for $i = 1, 2, \ldots, N$.

Each coding neuron $n_{C,i}$ receives excitatory signals from all the input neurons $n_{I,j}$ with a weight of $F_{ij} \geq 0$. That is, each coding neuron has a row of the matrix $F \in \mathbb{R}_{\geq 0}^{N \times M}$. In addition, neuron $n_{C,i}$ receives inhibitory signals from all other coding neurons $n_{C,j}$ with weight $-W_{ij} \leq 0$. $W$ denotes this matrix of weights: $W \in \mathbb{R}_{\geq 0}^{N \times N}$ and $\mathrm{diag}(W) = \mathbf{0}$. The firing thresholds are $\boldsymbol{\theta} = [\theta_1, \theta_2, \ldots, \theta_N]^T$ and the matrix $W + \Theta$, $\Theta = \mathrm{diag}(\boldsymbol{\theta})$, appears often and will denote it as $H \stackrel{\text{def}}{=} W + \Theta$. Each neuron $n_{C,i}$ also receives a constant negative bias of $-(1 - \gamma)\lambda_1 s_i$ where $0 \leq \gamma < 1$ is an important parameter that will be varied during the learning process to be detailed momentarily.

Each input neuron $n_{I,i}$, $i = 1, 2, \ldots, M$, with firing threshold fixed to be 1, receives a bias of $(1 - \gamma)x_i$. Typically $x_i$ corresponds to the $i$-th pixel value of an input image in question during the learning process. In addition, it receives excitatory spikes from each of the coding neurons with weights $\gamma B_{ij} \geq 0$. That is each input neuron has a row of the matrix $B \in \mathbb{R}_{\geq 0}^{M \times N}$. These excitatory signals from the coding neurons constitute the crucial feedback mechanism we devised here that enables dictionary learning.

## B   PROOF OF THEOREMS

### B.1   THEOREM 1: SNN DYNAMICS, TRAJECTORY, AND LIMIT POINTS

In the simplest case when none of the neurons are inter-connected and $\rho_i(0) < \theta_i$ for all $i$, then $\mu_i(t) = \beta_i$ for all $i$ and all $t \geq 0$. Hence those neurons $n_i$ with $\beta_i > 0$ produces a spike train of constant inter-spike interval of $\theta_i / \beta_i$; those neurons with $\beta_i \leq 0$ will have no spiking activities. When however the neurons are inter-connected, the dynamics becomes non-trivial. It turns out that one can so describe the dynamics mathematically that useful properties related to the current and spike train can be derived. Consequently, a network of spiking neurons can be configured to help solve certain practical problems.

Given a system of $N$ neurons $n_i$, $i = 1, 2, \ldots, N$, we use vector notations $\boldsymbol{\mu}(t)$ and $\boldsymbol{\sigma}(t)$ to denote the $N$ currents and spike trains. The vector $\boldsymbol{\beta}$ and $\boldsymbol{\theta}$ are the input biases and firing thresholds. The convolution $(\alpha * \boldsymbol{\sigma})(t)$ is the $N$-vector whose $i$-th component is $(\alpha * \sigma_i)(t)$. For simplicity, we consider only $\tau = 1$ throughout the paper. Thus $\alpha(t) = e^{-t}$ for $t \geq 0$ and 0 otherwise. Equation 1 in vector form is

$$\boldsymbol{\mu}(t) = \boldsymbol{\beta} + W(\alpha * \boldsymbol{\sigma})(t) \tag{13}$$

where $W \in \mathbb{R}^{N \times N}$ and $W_{ii} = 0$, encodes the inhibitory/excitatory connections among the neurons. Because $\frac{d}{dt}(\alpha * \sigma)(t) = \sigma(t) - (\alpha * \sigma)(t)$, we have

$$\dot{\boldsymbol{\mu}}(t) = \boldsymbol{\beta} - \boldsymbol{\mu}(t) + W \cdot \boldsymbol{\sigma}(t). \tag{14}$$

Filtering Equation 14 yields

$$\dot{\mathbf{u}}(t) = \boldsymbol{\beta} - \mathbf{u}(t) + W \mathbf{a}(t) + (\boldsymbol{\mu}(0) - \mathbf{u}(t))/t$$
$$\mathbf{u}(t) - \Theta \mathbf{a}(t) = \boldsymbol{\beta} + (W - \Theta) \mathbf{a}(t)$$
$$+ (\boldsymbol{\mu}(0) - \mathbf{u}(t))/t - \dot{\mathbf{u}}(t) \tag{15}$$

where $\Theta = \mathrm{diag}(\boldsymbol{\theta})$. Theorem B.1 has been established previously in Tang et al. (2017) in a slightly different form. We attach the proof consistent to our notations below for completeness. It is established under the following assumptions:

- The currents of all neurons remain bounded from above, $\|\boldsymbol{\mu}(t)\|_\infty \leq B$ for all $t \geq 0$ for some $B > 0$. This implies no neuron can spike arbitrarily fast, and the fact that neurons cannot spike arbitrarily rapidly implies the currents are bounded from below as well

- There is a positive number $r > 0$ such that whenever the numbers $t_{i,k}$ and $t_{i,k+1}$ exist, $t_{i,k+1} - t_{i,k} \leq 1/r$. This assumption says that unless a neuron stop spiking althogether after a certain time, the duration between consecutive spike cannot become arbitrarily long.

**Theorem B.1.** *As $t \to \infty$, $\dot{\mathbf{u}}(t)$, $\frac{1}{t}(\boldsymbol{\mu}(0) - \mathbf{u}(t))$ and $\max(\mathbf{u}(t), \mathbf{0}) - \Theta\,\mathbf{a}(t)$ all converge to $\mathbf{0}$.*

**Proof.** *Let*

$$\mathcal{A} = \{\, i \mid \text{neuron-}i \text{ spikes infinitely often} \,\}$$

*($\mathcal{A}$ stands for "active"), and*

$$\mathcal{I} = \{\, i \mid \text{neuron-}i \text{ stop spiking after a finite time} \,\}$$

*($\mathcal{I}$ stands for "inactive"). First consider $i \in \mathcal{I}$. Let $t_{i,k}$ be the time of the final spike. For any $t > t_{i,k}$,*

$$
\begin{aligned}
u_i(t) &= \frac{1}{t}\int_0^{t_{i,k}} \mu_i(s)\,ds + \frac{1}{t}\int_{t_{i,k}}^{t} \mu_i(s)\,ds \\
&= \frac{1}{t}\int_0^{t_{i,k}} \mu_i(s)\,ds + \frac{1}{t}\rho_i(t) \\
&= \theta_i a_i(t) + \frac{1}{t}\rho_i(t)
\end{aligned}
$$

*Note that $\rho_i(t) \leq \theta_i$ always. If $\rho_i(t) \geq 0$, then*

$$0 \leq \max(u_i(t),\, 0) - \theta_i a_i(t) \leq \theta_i/t.$$

*If $\rho_i(t) < 0$,*

$$-\theta_i a_i(t) \leq \max(u_i(t),\, 0) - \theta_i a_i(t) \leq 0.$$

*Since $i \in \mathcal{I}$, $a_i(t) \to 0$ obviously. Thus*

$$\max(u_i(t),\, 0) - \theta_i a_i(t) \to 0.$$

*Consider the case of $i \in \mathcal{A}$. For any $t > 0$, let $t_{i,k}$ be the largest spike time that is no bigger than $t$. Because $i \in \mathcal{A}$, $t_{i,k} \to \infty$ as $t \to \infty$.*

$$
\begin{aligned}
u_i(t) &= \frac{1}{t}\int_0^{t_{i,k}} \mu_i(s)\,ds + \frac{1}{t}\int_{t_{i,k}}^{t} \mu_i(s)\,ds \\
&= \theta_i a_i(t) + \frac{1}{t}\int_{t_{i,k}}^{t} \mu_i(s)\,ds.
\end{aligned}
$$

*Furthermore, note that because of the assumption $t_{i,k+1} - t_{i,k} \leq 1/r$ always, where $r > 0$, $\liminf a_i(t) \geq r$. In otherwords, there is a time $T$ large enough such that $a_i(t) \geq r/2$ for all $i \in \mathcal{A}$ and $t \geq T$. Moreover, $0 \leq t - t_{i,k} \leq t_{i,k+1} - t_{i,k} \leq 1/r$ and $\mu_i(t) \in [B_-, B_+]$. Thus*

$$\frac{1}{t}\int_{t_{i,k}}^{t} \mu_i(s)\,ds \in \frac{1}{t}[B_-, B_+]/r \to 0.$$

*When this term is eventually smaller in magnitude than $\theta_i a_i(t)$, we have*

$$u_i(t) - \theta_i a_i(t) \to 0.$$

*or equivalently,*

$$\max(u_i(t),\, 0) - \theta_i a_i(t) \to 0.$$

∎

Applying Theorem B.1 to Equation 15 yields the following.

**Theorem B.2.** *Given any $\epsilon > 0$, there exists $T > 0$ such that for all $t > T$,*

$$\|(\mathbf{u}(t) - \Theta\,\mathbf{a}(t)) - (\boldsymbol{\beta} + (W - \Theta)\,\mathbf{a}(t))\|_\infty < \epsilon.$$

The following theorem characterizes limit points of the trajectory $(\mathbf{u}(t), \mathbf{a}(t))$. Recall that $(\mathbf{u}^*, \mathbf{a}^*)$ is a limit point if given any $\epsilon > 0$, there exists a time $T > 0$ large enough such that $(\mathbf{u}(T), \mathbf{a}(T))$ is within $\epsilon$ to $(\mathbf{u}^*, \mathbf{a}^*)$.

**Theorem B.3.** *Given any limit point $(\mathbf{u}^*, \mathbf{a}^*)$, we must have $\boldsymbol{\beta} + (W - \Theta)\,\mathbf{a}^* \leq \mathbf{0}$, $\mathbf{a}^* \odot (\boldsymbol{\beta} + (W - \Theta)\,\mathbf{a}^*) = \mathbf{0}$ and $\mathbf{a}^* \geq \mathbf{0}$, where $\odot$ is the elementwise product.*

**Proof.** *Theorem B.1 shows that $\mathbf{u}^* - \Theta\,\mathbf{a}^* \leq \mathbf{0}$ and the elementwise product $\mathbf{a}^* \odot (\mathbf{u}^* - \Theta\,\mathbf{a}^*) = \mathbf{0}$. But Theorem B.2 shows that $\mathbf{u}^* - \Theta\,\mathbf{a}^* = \boldsymbol{\beta} + (W - \Theta)\,\mathbf{a}^*$ and the theorem here is established.* ∎

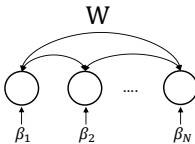

Figure 5: A 1-layer LCA network for sparse coding.

## B.2 Non-negative sparse coding by spiking neural networks

Given a non-negative dictionary $D \in \mathbb{R}^{M \times N}_{\geq 0}$, a positive scaling vector $\mathbf{s} = [s_1, s_2, \ldots, s_N]^T \in \mathbb{R}^N_{>0}$ and an image $\mathbf{x} \in \mathbb{R}^M_{\geq 0}$, the non-negative sparse coding problem can be formulated as

$$\mathbf{a}^* = \arg\min_{\mathbf{a} \geq \mathbf{0}} l(\mathbf{a}), l(\mathbf{a}) = \frac{1}{2}\|\mathbf{x} - D\mathbf{a}\|_2^2 + \lambda\|S\mathbf{a}\|_1 \tag{16}$$

where $S = \mathrm{diag}(\mathbf{s})$. Using the well-known KKT condition in optimization theory, see for example Boyd & Vandenberghe (2004), $\mathbf{a}^*$ is an optimal solution iff there exists $\mathbf{e}^* \in \mathbb{R}^N$ such that all of the following hold:

$$\begin{cases} \mathbf{0} \in \partial l(\mathbf{a}^*) - \mathbf{e}^* & \text{(stationarity)} \\ \mathbf{e}^* \odot \mathbf{a}^* = \mathbf{0} & \text{(complementarity)} \\ \mathbf{a}^* \geq \mathbf{0}, \mathbf{e}^* \geq \mathbf{0} & \text{(feasibility)} \end{cases} \tag{17}$$

where $\partial l$ is the generalized gradient of $l$. Note that the generalized gradient $\partial l(\mathbf{a})$ is $D^T D\mathbf{a} - D^T\mathbf{x} + \lambda\mathbf{s} \odot \partial\|\mathbf{a}\|_1$ and that $\partial|a_i| = 1$ when $a_i > 0$ and equals the interval $[-1, 1]$ when $a_i = 0$. Straightforward derivation then shows that $\mathbf{a}^*$ is an optimal solution iff

$$D^T\mathbf{x} - \lambda\mathbf{s} - D^T D\mathbf{a}^* \leq \mathbf{0} \text{ ; and}$$
$$\mathbf{a}^* \odot (D^T\mathbf{x} - \lambda\mathbf{s} - D^T D\mathbf{a}^*) = \mathbf{0}. \tag{18}$$

We now configure a $N$-neuron system depicted in Figure 5 so as to solve Equation 16. Set $\boldsymbol{\theta} = \mathrm{diag}(D^T D)$ as the firing thresholds and set $\boldsymbol{\beta} = D^T\mathbf{x} - \lambda\mathbf{s}$ as the bias. Define the inhibition matrix to be $-(D^T D - \Theta), \Theta = \mathrm{diag}(\boldsymbol{\theta})$. Thus neuron-$j$ inhibits neuron-$i$ with weight $-\mathbf{d}_i^T\mathbf{d}_j \leq 0$. In this configuration, it is easy to establish $\|\boldsymbol{\mu}(t)\|_\infty \leq C$ for all $t \geq 0$ for some $C > 0$ as all connections are inhibitions. From Theorem B.3, any limit point $(\mathbf{u}^*, \mathbf{a}^*)$ of the trajectory $(\mathbf{u}(t), \mathbf{a}(t))$ satisfies $\boldsymbol{\beta} + (W - \Theta)\mathbf{a}^* \leq \mathbf{0}$ and $\mathbf{a}^* \odot (\boldsymbol{\beta} + (W - \Theta)\mathbf{a}^*) = \mathbf{0}$. But $\boldsymbol{\beta} = D^T\mathbf{x} - \lambda\mathbf{s}$ and $W - \Theta = -D^T D$. Thus $\mathbf{a}^*$ solves Equation 16. And in particular, if the solution to Equation 16 is unique, the trajectory can only have one limit point, which means in fact the trajectory converges to the sparse coding solution. This result can be easily extended to the network in Figure 1(a) by expanding the bias into another layer of input neuron with $F = D^T$.

## B.3 Theorem 2: sparse coding with feedback perturbation

Equation 5, reflecting the structure of the coding and input neurons, takes the form:

$$\begin{bmatrix} \mathbf{e}_\gamma(t) \\ \mathbf{f}_\gamma(t) \end{bmatrix} \stackrel{\text{def}}{=} \begin{bmatrix} \mathbf{u}_\gamma(t) - \Theta\mathbf{a}_\gamma(t) \\ \mathbf{v}_\gamma(t) - \mathbf{b}_\gamma(t) \end{bmatrix} = \begin{bmatrix} -(1-\gamma)\lambda_1\mathbf{s} \\ (1-\gamma)\mathbf{x} \end{bmatrix} + \begin{bmatrix} -H & F \\ \gamma B & -I \end{bmatrix} \begin{bmatrix} \mathbf{a}_\gamma(t) \\ \mathbf{b}_\gamma(t) \end{bmatrix} + \boldsymbol{\Delta}(t) \tag{19}$$

$(\mathbf{u}(t), \mathbf{v}(t))$ and $(\mathbf{a}(t), \mathbf{b}(t))$ denote the average currents and spike rates for the coding and input neurons, respectively, and $H \stackrel{\text{def}}{=} W + \Theta$. Note that $\max(\mathbf{u}_\gamma(t), \mathbf{0}) - \Theta\mathbf{a}_\gamma(t), \max(\mathbf{v}_\gamma(t), \mathbf{0}) - \mathbf{b}_\gamma(t)$ and $\boldsymbol{\Delta}(t)$ all converge to $\mathbf{0}$ as $t \to \infty$.

**Theorem B.4.** *Consider the configuration $F^T = B = D$ and $\gamma \in [0, 1)$. Suppose the soma currents and thus spike rates $\|\mathbf{a}_\gamma(t)\|_\infty$ are bounded. Let $H = D^T D + (\lambda_1\gamma^c)^{-1}\Delta_H, \gamma^c = 1 - \gamma$, be such that $4\|\Delta_H\|_1\|\mathbf{a}_\gamma(t)\|_\infty < \min\{s_i\}$. Then, for any $\epsilon > 0$ there is $T > 0$ such that for all $t > T$, $\|\mathbf{a}_\gamma(t) - \hat{\mathbf{a}}(t)\|_\infty < \epsilon$ and $\hat{\mathbf{a}}(t)$ solves Equation 16 with $S$ replaced by $\hat{S}$ where $\|S - \hat{S}\|_\infty < \min\{s_i\}/2$.*

**Proof.** *Consider $\tau > 0$ and define the vectors $\hat{\mathbf{a}}(t)$ and $\hat{\mathbf{u}}(t)$ for $t \geq 0$ by each of their components:*

$$(\hat{a}_i(t), \hat{u}_i(t)) = \begin{cases} (a_{\gamma,i}(t), \theta_i a_{\gamma,i}(t)) & \text{if } a_{\gamma,i}(t) \geq \tau, \\ (0, \quad \min(u_{\gamma,i}(t), 0)) & \text{otherwise,} \end{cases}$$

*where $\boldsymbol{\theta}$ is the diagonal of $D^T D$. Denote the perturbations $\boldsymbol{\Delta}_a(t) \overset{\text{def}}{=} \hat{\mathbf{a}}(t) - \mathbf{a}_\gamma(t)$, $\boldsymbol{\Delta}_u(t) \overset{\text{def}}{=} \hat{\mathbf{u}}(t) - \mathbf{u}_\gamma(t)$, $\hat{\mathbf{e}}_\gamma(t) \overset{\text{def}}{=} \hat{\mathbf{u}}(t) - \Theta\hat{\mathbf{a}}(t)$, and $\boldsymbol{\Delta}_e(t) \overset{\text{def}}{=} \hat{\mathbf{e}}(t) - \mathbf{e}_\gamma(t)$. This construction of $\hat{\mathbf{u}}(t)$ and $\hat{\mathbf{a}}(t)$ ensures $\|\boldsymbol{\Delta}_a(t)\|_\infty < \epsilon$, $\hat{\mathbf{e}}(t) \leq \mathbf{0}$, and $\hat{\mathbf{e}}(t) \odot \hat{\mathbf{a}}(t) = \mathbf{0}$. Recall that $\max(\mathbf{u}_\gamma(t), \mathbf{0}) - \Theta\mathbf{a}_\gamma(t) \to \mathbf{0}$ (Theorem B.1); thus $\|\boldsymbol{\Delta}_u(t)\|_\infty < 2\tau$ at $t$ large enough.*

*Next, observe that $\mathbf{v}_\gamma(t) \geq \mathbf{0}$ always $\boldsymbol{\nu}_\gamma(t) \geq \mathbf{0}$ always, for any setting $\gamma$ in $[0, 1)$. Thus Theorem B.1 implies*

$$\mathbf{b}_\gamma(t) - [(1 - \gamma)\mathbf{x} + \gamma B\mathbf{a}_\gamma(t)] \to \mathbf{0} \tag{20}$$

*as $t \to \infty$. From Equation 19, this implies that*

$$\boldsymbol{e}_\gamma(t) = \gamma^c(D^T\mathbf{x} - \lambda_1\mathbf{s} - D^T D\mathbf{a}_\gamma(t) - \lambda_1\Delta_H\mathbf{a}_\gamma(t) + \boldsymbol{\Delta}(t))$$

*for some $\boldsymbol{\Delta}(t)$ where $\|\boldsymbol{\Delta}(t)\|_\infty \to 0$. Thus*

$$(\gamma^c)^{-1}\hat{\mathbf{e}}(t) = D^T\mathbf{x} - \lambda_1\hat{\mathbf{s}} - D^T D\hat{\mathbf{a}}(t)$$

*where $\hat{\mathbf{s}} = \mathbf{s} - (\boldsymbol{\eta}(t) + \boldsymbol{\zeta}(t))$, $\boldsymbol{\eta}(t) = \Delta_H\mathbf{a}_\gamma(t)$ and $\boldsymbol{\zeta}(t) = \lambda_1^{-1}(D^T D\boldsymbol{\Delta}_a(t) + \boldsymbol{\Delta}_e(t)/\gamma^c + \boldsymbol{\Delta}(t))$. By assumption on $\Delta_H$, $\mathbf{s} - \boldsymbol{\eta}(t) > (3/4)\mathbf{s} > \mathbf{0}$. Moreover, $\|\boldsymbol{\zeta}(t)\|_\infty$ can be made arbitrarily small by taking $t$ and $1/\tau$ large enough. Thus there exist $\tau, T > 0$ such that for all $t > T$, $\|\boldsymbol{\Delta}_a(t)\|_\infty < \epsilon$ and $\|\hat{\mathbf{s}}(t) - \mathbf{s}\|_\infty < \min\{s_i\}/2$, implying in particular $\hat{\mathbf{s}}(t) > \mathbf{s}/2 > \mathbf{0}$. Finally, note that*

$$\hat{\mathbf{a}}(t) \geq \mathbf{0}, \quad (\gamma^c)^{-1}\hat{\mathbf{e}}(t) \leq \mathbf{0}, \quad (\gamma^c)^{-1}\hat{\mathbf{e}}(t) \odot \hat{\mathbf{a}}(t) = \mathbf{0},$$

*which shows (recall Equation 18) that $\hat{\mathbf{a}}(t)$ solves Equation 16 with $S$ replaced by $\hat{S}$ and the proof is complete.* ∎

At present, we cannot establish a priori that the currents stay bounded when $\gamma > 0$. Nevertheless, the theorem is applicable in practice as long as the observed currents stay bounded by some $C$ for $0 \leq t \leq T$ and $C/T$ is small enough. See Section 3.3 for further comments.

### B.4 THEOREM 3: GRADIENT CALCULATIONS FROM CONTRASTIVE LEARNING

**Theorem B.5.** *Given any $\epsilon > 0$, there is a $T > 0$ such that for all $t, t' > T$,*

$$\|\kappa(B\mathbf{a}_\kappa(t') - \mathbf{x}) - (\mathbf{b}_\kappa(t') - \mathbf{b}_0(t))\|_\infty < \epsilon, \tag{21}$$

$$\|\kappa^c H(\mathbf{a}_0(t) - \mathbf{a}_\kappa(t')) - \kappa(H - FB)\mathbf{a}_\kappa(t') + (\kappa^c\mathbf{e}_0(t) - \mathbf{e}_\kappa(t'))\|_\infty < \epsilon. \tag{22}$$

**Proof.** *Equation 20 implies that*

$$\kappa(B\mathbf{a}_\kappa(t') - \mathbf{x}) - (\mathbf{b}_\kappa(t') - \mathbf{b}_0(t)) \to \mathbf{0} \quad \text{as } t, t' \to \infty,$$

*establishing Equation 21. From Equations 19 and 20*

$$-\kappa^c\lambda\mathbf{s} - \kappa^c H\mathbf{a}_0(t) + \kappa^c F\mathbf{x} - \kappa^c\mathbf{e}_0(t) \to \mathbf{0}, \text{ and,}$$

$$-\kappa^c\lambda\mathbf{s} - H\mathbf{a}_\kappa(t) + \kappa^c F\mathbf{x} + \kappa FB\mathbf{a}_\kappa(t) - \mathbf{e}_\kappa(t) \to \mathbf{0}.$$

*Equation 22 thus follows.* ∎

## C COMPARISONS WITH PRIOR WORK

### C.1 COMPARISONS OF DYNAMICAL NEURAL NETWORKS

Table 1 provides a summary of the development of three types of dynamical neural networks: Hopfield network, Boltzmann machine, and sparse coding network.

| Hopfield network | |
|---|---|
| Neuron model | Binary or continuous (Hopfield, 1982; 1984) |
| Activation | Binary: Thresholding
Continuous: Any bounded, differentiable, strictly increasing function |
| Topology | Arbitrary symmetric bidirectional connections |
| Learning | Binary: Hebbian rule
Continuous: contrastive learning (Movellan, 1990) |
| Limit point | Many local minimum |
| Usage | Associative memory, constraint satisfaction problem |

| Boltzmann machine | |
|---|---|
| Neuron model | Binary or continuous (for visible units) (Ackley et al., 1985; Freund & Haussler, 1992) |
| Activation | Logistic |
| Topology | BM: Arbitrary symmetric bidirectional connections
RBM:Two-layer with symmetric forward/backward (Hinton et al., 2006) |
| Learning | BM: contrastive learning
RBM: contrastive divergence |
| Limit point | Many local minimum |
| Usage | Generative model, constraint satisfaction problem |

| Sparse coding network | |
|---|---|
| Neuron model | Continuous or spiking (Rozell et al., 2008; Shapero et al., 2014) |
| Activation | Rectified linear |
| Topology | Two-layer with feedforward, lateral, and **feedback** connections |
| Learning | **Contrastive learning with weight consistency** |
| Limit point | Likely unique (Bruckstein et al., 2008) |
| Usage | Representation learning with sparse prior, image denoising and super-resolution, compressive sensing |

Table 1: Comparison between dynamical neural networks. Text in boldface indicates the new results established in this work.

## C.2 COMPARISONS OF DICTIONARY LEARNING NETWORKS

As we discussed in Section 1.1, there are several prior work that qualitatively demonstrate dictionary learning in dynamical neural networks. The prior work Földiak (1990); Zylberberg et al. (2011); Brito & Gerstner (2016); Hu et al. (2014); Seung & Zung (2017); Vertechi et al. (2014); Brendel et al. (2017) employ a feedforward-only network topology as shown in Figure 1(a), and are unable to compute the true gradient for dictionary learning from local information. These work hence rely on additional heuristic or assumptions on input data for learning to work. In contrast, we propose to introduce feedback connections as shown in Figure 1(b), which allows us to solve the fundamental problem of estimating the true gradient. Recall the dictionary learning objective function (Equation 6 in the main text)

$$\underset{\mathbf{a}^{(p)} \geq \mathbf{0}, D \geq \mathbf{0}}{\arg\min} \sum_{p=1}^{P} l(D, \mathbf{x}^{(p)}, \mathbf{a}^{(p)}), \quad l(D, \mathbf{x}, \mathbf{a}) = \frac{1}{2}\|\mathbf{x} - D\mathbf{a}\|_2^2 + \lambda_1\|S\mathbf{a}\|_1 + \frac{\lambda_2}{2}\|D\|_F^2, \quad (23)$$

and the true stochastic gradient of the learning problem is (Equation 7 in the main text)

$$D^{(\text{new})} \leftarrow D - \eta \left( (D\mathbf{a} - \mathbf{x})\mathbf{a}^T + \lambda_2 D \right). \quad (24)$$

Here we provide a detailed discussion on the difference and limitations of prior work.

The first line of work is the so-called Hebbian/anti-Hebbian network Földiak (1990); Zylberberg et al. (2011); Brito & Gerstner (2016); Hu et al. (2014); Seung & Zung (2017). The principle of learning in these work is to apply Hebbian rules to learn excitatory feedforward weights (strengthen an excitatory connection if the two connected neurons, input and coding neurons, have strong activations) and anti-Hebbian learning for inhibitory lateral weights (strengthen an inhibitory connection if the two connected neurons, both coding neurons, have strong activations). Due to the heuristic nature of this learning strategy, it is unclear whether this approach can solve the dictionary learning problem in Equation 23. Zylberberg et al. (2011) argues that the Hebbian rule can approximate the gradient of the reconstruction error term under a strong assumption that for some batch of successive inputs, the activities of the coding neurons are uncorrelated (i.e., their computed sparse codes are uncorrelated), and all the neurons have the same average activations. However, since the anti-Hebbian rule cannot ensure weight consistency between feedforward and lateral weights, the sparse code computed by the network may not correspond to any quantity related to the dictionary learning objective function. A significant gap still exists between this learning strategy and a rigorous learning objective function. Hu et al. (2014); Pehlevan et al. (2018) argues that this learning framework arises from a different objective function other than Equation 23. By using the following "similarity matching" objective function,

$$\arg \min_A \|X^T X - A^T A\|_F^2, \tag{25}$$

where $X \in \mathbb{R}^{M \times P}$ and $A \in \mathbb{R}^{N \times P}$ are formed by stacking the input $\mathbf{x}$ and the sparse codes $\mathbf{a}$ along the columns, respectively, they are able to derive learning rules that resemble the principles of the Hebbian/anti-Hebbian learning heuristic. This formulation is somewhat different from the dictionary learning objective function we are interested in.

The second line of work Vertechi et al. (2014); Brendel et al. (2017) proposes to learn the lateral weights according to the feedforward weights instead of using anti-Hebbian rules to address global weight consistency, although the learning of feedforward weight still follows Hebbian rules, giving the following update equation,

$$D^{(\text{new})} \leftarrow D + \eta \left( \mathbf{x}\mathbf{a}^T - \lambda_2 D \right). \tag{26}$$

It can be seen that Equation 26 is not an unbiased estimate of the true stochastic gradient in Equation 24. Hence in theory the convergence of learning to an optimal solution cannot be guaranteed. Nontheless, Vertechi et al. (2014); Brendel et al. (2017) show that under the assumption that the input is whitened and centered, empirically Equation 26 can progressively learn a dictionary with improved reconstruction performance. For the general case of non-whitened input, the authors proposed a modified dictionary learning objective function:

$$l = \frac{1}{2}(\mathbf{x}_c - D\mathbf{a})^T C^{-1}(\mathbf{x}_c - D\mathbf{a}) + \lambda_1 \|\mathbf{a}\|_1 \tag{27}$$

where $\mathbf{x}_c$ is the mean-removed training sample and $C$ is the covariance, different from the common dictionary learning objective that we are interested in Equation 23. Note that in Vertechi et al. (2014); Brendel et al. (2017), it is unclear whether the non-negative constraints $D \geq 0, \mathbf{a} \geq 0$ exist in the learning objective, although fundamentally the two constraints cannot be omitted: $D \geq 0$ is needed so that the input neurons are only excitatory; $\mathbf{a} \geq 0$ is needed because spike rates cannot go negative. Once these two constraints are incorporated, the objective function in Equation 27 may not be very meaningful: the input to be estimated $\mathbf{x}_c$ may have negative entries, while the estimation $D\mathbf{a}$ is always non-negative.

In this work, we directly estimate the true stochastic gradient for dictionary learning, and therefore we do not need to make additional assumptions on the training input. As discussed in the main text, obtaining such estimate requires adding the feedback connections with the resulting non-trivial network dynamics. We provide extensive analysis and proofs and show that dictionary learning can be solved under this setting.

Finally, we note that the need for feedback has been repeatedly pointed out in training autoencoder networks (Hinton & McClelland, 1988; Burbank, 2015). Autoencoder networks do not have the lateral connections as presented in the sparse coding network. Reconstruction errors there are computed by running the network, alternating between a forward-only and a backward-only phase. In contrast, we compute reconstruction errors by having our network evolve simultaneously with

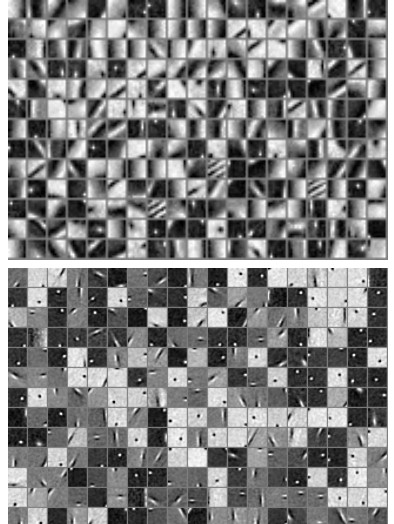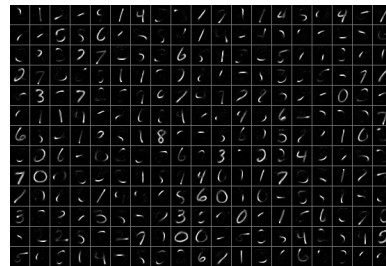

Figure 6: The figure shows a random subset of the dictionaries learned in spiking networks. They show the expected patterns of edges and textures (Lena), strokes and parts of the digits (MNIST), and Gabor-like oriented filters (natural scenes), similar to those reported in prior works (Rubinstein et al., 2010; Ranzato et al., 2007; Hoyer, 2004).

both feedforward and feedback signals tightly coupled together. Nevertheless, these models do not form strong back-coupled dynamical neural networks. Instead, they rely on staged processing much similar to a concatenation of feedforward networks. For our network, the dictionary learning relies only on locations of the dynamics' trajectories at large time which need not be close to a stable limit point. Simple computations between these locations that corresponding to two different network configurations yield the necessary quantities such as reconstruction error or gradients for minimizing a dictionary learning objective function.

# D    ADDITIONAL NUMERICAL EXPERIMENT RESULTS

## D.1    VISUALIZATION OF LEARNED DICTIONARIES

In Section 4, we presented the convergence of dictionary learning by dynamical neural networks on three datasets: Lena, MNIST, and SparseNet. Figure 6 shows the visualization of the respectively learned dictionaries. Unsurprisingly, these are qualitatively similar to the well-known results from solving dictionary learning using canonical numerical techniques.

## D.2    IMAGE DENOISING USING LEARNED DICTIONARIES

Here we further demonstrate the applicability of the dictionary learned by our dynamical neural networks. We use the dictionary learned from Dataset A (the Lena image) for a denoising task using a simple procedure similar to Elad & Aharon (2006): First we extract $8 \times 8$ overlapping patches from the noisy $512 \times 512$ Lena image generated with Gaussian noise. We then solve for the sparse coefficients of each patch in the non-negative sparse coding problem. Using the sparse coefficients, we can reconstruct the denoised patches, and a denoised image can be obtained by properly aligning and averaging these patches. On average, each patch is represented by only 5.9 non-zero sparse coefficients. Figure 7 shows a comparison between the noisy and the denoised image.

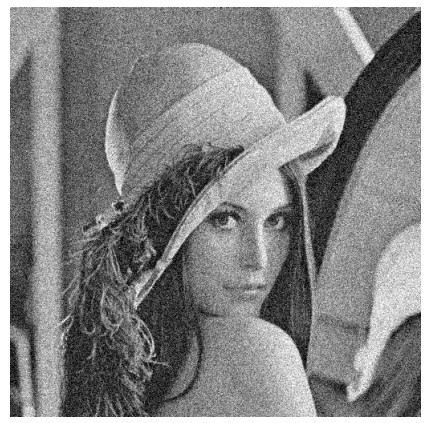 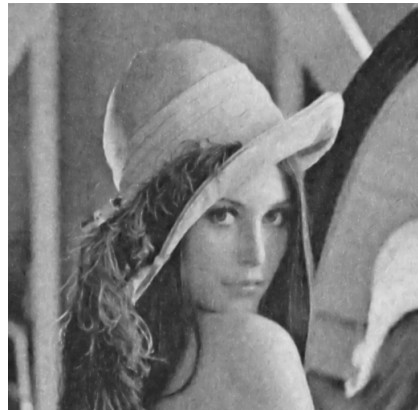

(a) Noisy image (PSNR=18.69dB)    (b) Denoised image (PSNR=29.31dB)

Figure 7: Image denoising using learned dictionary.

# E RELATIONSHIPS BETWEEN CONTINUOUS AND SPIKING NEURON MODEL FOR SPARSE CODING

Although in this work we focus our discussions and analysis on spiking neurons, the learning strategy and mechanism can be applied to networks with continuous-valued neurons. The close relationships between using spiking and continuous-valued neurons to solve sparse approximation problems has been discussed by Shapero et al. (2014); Tang et al. (2017). Here we attempt to provide an informal discussion on the connections between the two neuron models.

Following the derivation in Section 3, the dynamics of the spiking networks can be described using the average current and spike rates.

$$\dot{\mathbf{u}}(t) = \boldsymbol{\beta} - \mathbf{u}(t) + W\,\mathbf{a}(t) + (\boldsymbol{\mu}(0) - \mathbf{u}(t))/t \tag{28}$$

where $\mathbf{u}(t)$ and $\mathbf{a}(t)$ can be related by Theorem 1 as an "activation function".

$$\mathbf{a}(t) = \Theta^{-1}\max(\mathbf{u}(t), \mathbf{0}) + \boldsymbol{\Delta}(t), \quad \boldsymbol{\Delta}(t) \to 0 \tag{29}$$

Equation 28 and 29 are closely related to the dynamics of a network of continuous-valued neuron Rozell et al. (2008).

$$\dot{\mathbf{u}_{\mathbf{c}}}(t) = \boldsymbol{\beta_c} - \mathbf{u_c}(t) + W\mathbf{a_c}(t) \tag{30}$$

$$\mathbf{a_c}(t) = \max(\mathbf{u_c}(t), 0) \tag{31}$$

where $\mathbf{u_c}(t)$ is the internal state variable of each neuron, $\mathbf{a_c}(t)$ is the continuous activation value of each neuron, $\boldsymbol{\beta_c}$ is the input to each neuron, and $W$ is the connection weight between neurons. One can immediately see the similarity. Note that although such "ReLU" type, asymmetric activation function was not discussed in Rozell et al. (2008), it was later shown in Tang (2016) that this network dynamics can solve a non-negative sparse coding problem.

