# OpenReview forum: "Sparse Dictionary Learning by Dynamical Neural Networks"
_ICLR.cc/2019/Conference_

### Official Review · AnonReviewer1 · 2018-11-06
**An interesting, fully local learning approach to sparse coding**

**Rating:** 8
**Confidence:** 4

**Review:**

This paper proposes a dynamical neural network for sparse coding where all the interactions terms are learned.  In previous approaches (Rozell et al.) some weights were tied to the others.  Here the network consists of feedforward, lateral, and feedback weights, all of which have their own learning rule.  The authors show that the learned weights converge to the desired solution for solving the sparse coding objective.  This seems like a nice piece of work, an original approach that solves a problem that was never really fully resolved in previous work, and it brings things one step closer to both neurobiological plausibility and hardware implementation.

Other comments:

What exactly is being shown in Figure 2 is still not clear to me.

 It would be nice to see some other evaluations, for example sparsity vs. MSE tradeoff (this is reflected in the objective function in part but it would be nice to see the tradeoff).

There is recent work from Mitya Chklovskii's group on "similarity matching" that also addresses the problem of developing a fully local learning rule.  The authors should incorporate a discussion of this in their final paper.

---

> ### Author Response · Authors · 2018-11-18
> **Response**
>
> Figure 2 serves to illustrate our theoretical results and shows how the algorithm is run in practice. We revised the caption of Figure 2, providing a more detailed and clear description.
>
> We indeed cited and discussed the early "similarity matching" work (Hu et al. 2014) in our original submission. In our updated paper, we further included a later, more developed work (Pehlevan et al. 2018) in the reference. This line of work focuses on a novel learning objective function, while we study the sparse coding objective function that has been widely studied not only in neuroscience but also in signal processing and machine learning.

---

### Official Review · AnonReviewer3 · 2018-11-07
**Review of sparse dictionary learning by dynamical neural networks**

**Rating:** 9
**Confidence:** 4

**Review:**

The seminal work of Olshausen and Field on sparse coding is widely accepted as one of the main sources of inspiration for dictionary learning. This contribution makes the connection from dictionary learning back to a neuronal approach. Building on the Local Competitive Algorithm (LCA) of Rozell et al. and the theoretical analysis of Tang et al., this submission revisits the dictionary learning under two constraints that the gradient is learned locally and that the neural assemblies maintain consistent weight in the network. These constraints are relevant for a better understanding of the underlying principles in neuroscience and for applicative development on neuromorphic chipsets.

The proposed theorems extend the previous work of sparse coding with spiking neurons and address the update of dictionary using only information available from local neurons. The submission cares as well for the possible implementation on parallel architectures. The numerical experiments are conducted on three datasets and show the influence of weight initialization and the convergence on each dataset. An example of image denoising is provided in appendix.

---

### Official Review · AnonReviewer4 · 2018-11-12
**Local Sparse Codes**

**Rating:** 6
**Confidence:** 4

**Review:**

The authors study sparse coding models in which unit activations minimize a cost that combines: 1) the error between a linear generative model and the input data; an 2) the L1 norm of unit activations themselves. They seek models in which both the inference procedure -- generating unit activations in response to each input data example -- and the learning procedure -- updating network connections so that the inferences minimize the cost function -- are local. By "local" they mean that the update to each unit's activation, and the updates to the connection weights, rely only on information about the inputs and outputs from that unit / connection. In a biological neural network, these are the variables represented by pre- and post-synaptic action potentials and voltages, and in hardware implementations, operations on these variables can be performed without substantially coordinating between different parts of the chip, providing strong motivation for the locality constraint(s).

The authors achieve a local algorithm that approximately optimizes the sparse coding objective function by using feedback: they send the sparse coding units' activities "back" to the input layer through feedback connections. In the case where the feedback connection matrix is the transpose of the sparse coding dictionary matrix (D), the elementwise errors in the linear generative model (e.g., the non-local part of the sparse coding learning rule obtained by gradient descent) are represented by the difference between the inputs and this feedback to the input layer: that difference can be computed locally at the input units and then sent back to the coding layer to implement the updates. The feedback connections B are updated in another local process that keeps them symmetric with the feedforward weights: B= D= F^T throughout the learning process.

The authors provide several theorems showing that this setup approximately solves the sparse coding problem (again, using local information), and show via simulation that their setup shows similar evolution of the loss function during training, as does SGD on the sparse coding cost function.

I think that the paper presents a neat idea --  feedback connections are too often ignored in computational models of the nervous system, and correspondingly in machine learning. At the same time, I have some concerns about the novelty and the presentation. Those are described below:

1. The paper is unnecessarily hard to read, at least in part due to a lack of notational consistency. As just one example, with B=D, why use two different symbols for this matrix? This just makes is so that your reader needs to keep track mentally of which variable is actually which other variable, and that quickly becomes confusing. I strongly recommend choosing the simplest and most consistent notation that you can throughout the paper.

2. Other recent studies also showed that feedback connections can lead to local updates successfully training neural networks: three such papers are cited below. The first two papers do supervised learning, while the third does unsupervised learning. It would be helpful for the authors to explain the key points of novelty of their paper: application of these feedback connection ideas to sparse coding. Otherwise, readers may mistakenly get the impression that this work is the first to use feedback connections in training neural networks.

Guerguiev, J., Lillicrap, T.P. and Richards, B.A., 2017. Towards deep learning with segregated dendrites. ELife, 6, p.e22901.

Sacramento, J., Costa, R.P., Bengio, Y. and Senn, W., 2018. Dendritic cortical microcircuits approximate the backpropagation algorithm. arXiv preprint arXiv:1810.11393.

Federer, C. and Zylberberg, J., 2018. A self-organizing short-term dynamical memory network. Neural Networks.

3. Given that the performance gains of the locality (vs something like SparseNet) are given such emphasis in the paper, those should be shown from the numerical experiments. This could be quantified by runtime, or some other measure.

4. The discussion of prior work is a little misleading -- although I'm sure this is unintentional. For example, at the top of p. 3, it is mentioned that the previous local sparse coding models do not have rigorous learning objectives. But then the appendix describes the learning objectives, and the approximations, made in the prior work. I think that the introduction should have a more transparent discussion of what was, and was not, in the prior papers, and how the current work advances the field.

5. The paper -- and especially appendix C2 -- makes strong emphasis of the importance finding local implementations of true gradient descent, as opposed to the approximations made by prior authors. I'm not sure that's such a big deal, given that Lillicrap et al. showed nicely in the paper cited below that any learning rule that is within 90 degrees of true gradient descent will still minimize the cost function: even if an algorithm doesn't move down the steepest path, it can still have updates that always move "downhill", and hence minimize the loss function. Consequently, I think that some justification is needed showing that the current model, being closer to true gradient descent, really outperforms the previous ones.

Lillicrap, T.P., Cownden, D., Tweed, D.B. and Akerman, C.J., 2016. Random synaptic feedback weights support error backpropagation for deep learning. Nature communications, 7, p.13276.

---

> ### Author Response · Authors · 2018-11-18
> **Response**
>
> 1. Our intention to separate B and D is for their different physical meanings: the former corresponds to particular connection weights and the latter is an argument in an optimization problem. The reviewer's feedback to merge B and D certainly seems useful. We will further revise our paper with more succinct mathematical notation but need a little more time to ensure the presentation remains coherent.
>
> 2. We do not intend to claim to be the first to use feedback connections to train neural networks. This idea has a long history in contrastive learning (as we pointed out in the introduction) and can be even traced back to the recirculation algorithm for training autoencoders (Hinton \& McClelland, 1988) that we discussed in Appendix C.2. We have updated Section 1.2 to incorporate the growing body of work in this direction suggested by the reviewer, and help clarify our contributions.
>
> 3. The algorithm we established can be mapped to a massively parallel architecture and, in principle, lead to performance gains in two ways: the run time can be shorter due to a higher level of parallelism, and the energy cost of each operation can be lower because the memory is located closer to each computation unit. To further quantify the efficiency advantage of the algorithm, one must design a new parallel computer architecture such as (Davies et al., 2018), since the efficiency gain is coupled with the ability to build hardware differently. Otherwise, there is no easy comparison to make against a conventional numerical optimization algorithm implemented on a general-purpose CPU. The goal of this paper is to effect dictionary learning on this specialized computational model, which is already a non-trivial task, and we hope this work can motivate future study of hardware designs to realize the potential performance gains.
>
> 4. We have updated Section 1.1 and Appendix C.2 to avoid potential confusions pointed out by the reviewer. In summary, the prior work that we cited all have significant gaps between the learning rules and solving the dictionary learning objective functions. The arguments made by each work are detailed and discussed in Appendix C.2.
>
> 5. We believe our ability to compute the correct gradient is an important contribution, notwithstanding the reviewer's highly relevant comments. First, it is known to the deep learning and more generally the machine learning community that gradients need not be exact. For example, in the review article Optimization Methods for Large-Scale Machine Learning by Bottou/Curtis/Nocedal in SIAM Review, 2018, Equation 4.7a shows that as long as the approximate gradients on the average are uniformly smaller than 90 degrees, convergence can occur. In practice, however, it is hard to establish such a property in a non-trivial way. Such an acute angle property is satisfied by using an *exact* gradient on a batch of data points. In this case, the approximate gradients are unbiased estimates of the full-batch gradient and thus make a zero degree angle. In the works that we cited in our article that use approximations to gradient, none claim that the approximate gradients employed satisfy the acute angle property. This brings us to our second point. The work by Lillicrap et. al that the reviewer cited shows a number of excellent results. Note that, however, this work does not explicitly prove that the approximate gradient used satisfy the acute angle property. It shows that a random (but fixed) projection of error will lead the weights being so adjusted that the random projection becomes a suitable approximate gradient. Moreover, while successful learning is observed in quite a few examples, the authors established convergence proof in the supplemental material (Note 11) only for a linear network without nonlinear activations. As discussed in Note 16, the theoretical results are limited so far. We thus circumvent many theoretical difficulties by being able to compute exact gradients in the first place.

---

### Author Response · Authors · 2018-11-18
**Revision uploaded**

We thank all the reviewers for the detailed and constructive feedback. We have uploaded a revision of our submission to address the concerns as explained in the responses below.

---

### Meta-Review · Area_Chair1 · 2018-12-17
**Interesting work on how to train a dictionary from local information**

**Confidence:** 4
**Recommendation:** Accept (Poster)

**Metareview:**

While there has been lots of previous work on training dictionaries for sparse coding, this work tackles the problem of doing son in a purely local way. While previous work suggests that the exact computation of gradient addressed in the paper is not necessarily critical, as noted by reviewers, all reviewers agree that the work still makes important contributions through both its theoretical analyses and presented experiments. Authors are encouraged to work on improving clarity further and delineating their contribution more precisely with respect to previous results.